# Sustainable Synthesis of Cadmium Sulfide, with Applicability in Photocatalysis, Hydrogen Production, and as an Antibacterial Agent, Using Two Mechanochemical Protocols

**DOI:** 10.3390/nano12081250

**Published:** 2022-04-07

**Authors:** Zhandos Shalabayev, Matej Baláž, Natalya Khan, Yelmira Nurlan, Adrian Augustyniak, Nina Daneu, Batukhan Tatykayev, Erika Dutková, Gairat Burashev, Mariano Casas-Luna, Róbert Džunda, Radovan Bureš, Ladislav Čelko, Aleksandr Ilin, Mukhambetkali Burkitbayev

**Affiliations:** 1General and Inorganic Chemistry Department, Al-Farabi Kazakh National University, Al-Farabi Ave. 71, Almaty 050040, Kazakhstan; natalya.khan@kaznu.edu.kz (N.K.); nurlan_yelmira1@kaznu.edu.kz (Y.N.); batukhan.tatykaev@kaznu.kz (B.T.); kairat.burashev@mail.ru (G.B.); mukhambetkali.burkitbayev@kaznu.edu.kz (M.B.); 2Scientific Center for Anti-Infectious Drugs, Al-Farabi Ave. 75B, Almaty 050060, Kazakhstan; ilin_ai@mail.ru; 3Institute of Geotechnics, Slovak Academy of Sciences, Watsonova 45, 04001 Košice, Slovakia; balazm@saske.sk (M.B.); dutkova@saske.sk (E.D.); 4Chair of Building Materials and Construction Chemistry, Technische Universität Berlin, Gustav-Meyer-Allee 25, 13355 Berlin, Germany; adrian.augustyniak@zut.edu.pl; 5Faculty of Chemical Technology and Engineering, West Pomeranian University of Technology in Szczecin, Piastów Ave. 42, 71-065 Szczecin, Poland; 6Jožef Stefan Institute, Jamova Cesta 39, 01000 Ljubljana, Slovenia; nina.daneu@ijs.si; 7Central European Institute of Technology, Brno University of Technology, Purkynova 123, 612 00 Brno, Czech Republic; casas@vutbr.cz (M.C.-L.); ladislav.celko@ceitec.vcutbr.cz (L.Č.); 8Department of Physics of Materials, Charles University, 121 16 Prague, Czech Republic; 9Institute of Materials Research, Slovak Academy of Sciences, Watsonova 47, 04001 Košice, Slovakia; rdzunda@saske.sk (R.D.); rbures@saske.sk (R.B.)

**Keywords:** mechanosynthesis, combined milling, semiconductor, photocatalysis, hydrogen evolution, antibacterial activity, wastewater treatment

## Abstract

CdS nanoparticles were successfully synthesized using cadmium acetate and sodium sulfide as Cd and S precursors, respectively. The effect of using sodium thiosulfate as an additional sulfur precursor was also investigated (combined milling). The samples were characterized by XRD, Raman spectroscopy, XPS, UV-Vis spectroscopy, PL spectroscopy, DLS, and TEM. Photocatalytic activities of both CdS samples were compared. The photocatalytic activity of CdS, which is produced by combined milling, was superior to that of CdS, and was obtained by an acetate route in the degradation of Orange II under visible light irradiation. Better results for CdS prepared using a combined approach were also evidenced in photocatalytic experiments on hydrogen generation. The antibacterial potential of mechanochemically prepared CdS nanocrystals was also tested on reference strains of *E. coli* and *S. aureus*. Susceptibility tests included a 24-h toxicity test, a disk diffusion assay, and respiration monitoring. Bacterial growth was not completely inhibited by the presence of neither nanomaterial in the growth environment. However, the experiments have confirmed that the nanoparticles have some capability to inhibit bacterial growth during the logarithmic growth phase, with a more substantial effect coming from CdS nanoparticles prepared in the absence of sodium thiosulfate. The present research demonstrated the solvent-free, facile, and sustainable character of mechanochemical synthesis to produce semiconductor nanocrystals with multidisciplinary application.

## 1. Introduction

Recently, due to increasing environmental pollution, the requirements for searching for renewable energy sources to create environmentally friendly processes for decontaminating the environment are growing every day. Among all types of pollution, the most important is the discharge of wastewater from industrial enterprises [1]. The use of organic contaminants such as pesticides, herbicides, solvents, organic dyes, pharmaceuticals, cosmetics, etc., poses many problems and concerns for the scientific community and environmental regulators around the world [2]. To tackle this problem, photocatalysis is the best option among all methods, according to its simplicity and non-toxicity [3]. Consequently, the development of photocatalysts with enhanced activities for purifying wastewater from organic pollutants is hot now [4].

There are many types of photocatalysts, including metal oxides and ending metal sulfides [5,6]. Among them, photocatalysts based on CdS nanoparticles are of huge interest of scientists due to their wide bandgap energy (2.42 eV) [7] and reusability in photocatalytic processes [8]. In addition to photocatalysis, cadmium sulfide nanoparticles can be used in various fields of industries such as medicine [9], hydrogen production [10], photovoltaic devices [11], optoelectronics [12], diodes [13], gas sensors [14], etc. For example, in medicine, CdS nanoparticles reveal antibacterial [15], antimicrobial [16], and antifungal [17] activities against micro-organisms.

Recently, many methods have been developed to produce CdS nanoparticles such as microwave-assisted co-precipitation [18], the wet chemical method [19], microwave irradiation [20], reverse micellar methods [21], electrochemical deposition [22], the Langmuir–Blodgett method [23], solvothermal [24] and hydrothermal [25] processes, green synthesis [26], gamma irradiation [27], biological methods [28], and mechanochemical methods [29,30].

The last one is of particular interest in connection with the rapid supplementing of the high amount of mechanical energy to activate the chemical reaction between the reagents through the high-energy milling process [31]. This method is also environmentally friendly, since no harmful organic solvents are used [32]. Mechanochemistry is a universal tool for the synthesis of new substances, and it can be utilized in organic chemistry, supramolecular chemistry, organometallic compounds, polymers, inorganic chemistry [33], and even for waste treatment [34].

Until now, several papers reported on the synthesis of CdS nanoparticles by the mechanochemical approach. Acetate-route-based mechanosynthesis of nanocrystalline metal sulfides, in particular CdS nanoparticles, was proposed for the first time by the Slovak mechanochemists [35,36]. In this approach, the corresponding acetates and sodium sulfide were used as precursors. Later, using this approach, CdS nanoparticles with average sizes of 4–18 nm were produced in an industrial mill at ambient temperatures, in a very short processing time [29]. CdS nanoparticles also can be mechanochemically synthesized from elementary precursors. In the work [37], CdS nanoparticles with an average crystallite size of 8 nm were obtained.

In our previous work, needle-like copper sulfide (nCuS) nanocrystals have been synthesized, introducing in situ-prepared sulfur particles (using Na_2_S_2_O_3_ as an additional sulfur source and crystalline acid as a catalyst) to the mechanochemical reaction media of copper acetate monohydrate and sodium sulfide nonahydrate. Produced needle-like CuS nanocrystals showed less selective antibacterial activity in comparison with spherical ones [38]. In this work, we applied a similar approach for cadmium sulfide, with the aim to improve its application potential in photocatalytic degradation of organic dyes and antibacterial treatment. The novelty of this work is in the use of additional sulfur sources, rapid mechanochemical synthesis, and enhanced photocatalytic activity of CdS samples.

## 2. Materials and Methods

### 2.1. Chemicals

Sodium thiosulfate (Na_2_S_2_O_3_·5H_2_O, Sigma-Aldrich, Taufkirchen, Germany), citric acid (C_6_H_8_O_7_, Sigma-Aldrich, Taufkirchen, Germany), cadmium acetate (Cd(CH_3_COO)_2_·2H_2_O, Sigma-Aldrich, Taufkirchen, Germany), sodium sulfide (Na_2_S·9H_2_O, Acros Organics, Geel, Belgium), and Orange II sodium salt (C_16_H_11_N_2_NaO_4_S, Sigma-Aldrich, Taufkirchen, Germany) were of analytical grade and used without further purification. Distilled and deionized water was used in all washing and cleaning procedures.

### 2.2. Mechanochemical Synthesis of CdS Nanoparticles

In a typical mechanochemical synthesis procedure, the reactants (in an overall mass of 1 g) in stoichiometric amounts were transferred into a 100 mL silicon nitride (Si_3_N_4_) milling chamber containing 20 silicon nitride balls (10 mm), and they were milled for 5 min using an Activator 2SL planetary ball mill (Activator, Novosibirsk, Russia). The rotational speed of milling chambers was 550 rpm, the ball-to-powder ratio was 37, and all milling procedures were carried out under an air atmosphere.

CdS nanoparticles have been synthesized using two mechanochemical approaches: acetate route (aCdS) [35] and combined mechanochemical synthesis (cCdS), where sulfur prepared in situ should enter the formation of cadmium sulfide [38].

For the mechanochemical synthesis of aCdS nanoparticles by a simple acetate route, 0.526 g of Cd(CH_3_COO)_2_·2H_2_O and 0.474 g of Na_2_S·9H_2_O was milled in a stoichiometric amount according to the chemical reaction reported in [35]:(1)$$\mathrm{Cd}{\left({\mathrm{CH}}_{3}\mathrm{COO}\right)}_{2}\cdot 2H_{2}O+{\;\mathrm{Na}}_{2}S\cdot 9H_{2}O\;\to \mathrm{CdS}+2{\mathrm{CH}}_{3}\mathrm{COONa}+11H_{2}O$$

cCdS nanocrystals were prepared by the combined mechanosynthesis approach via co-milling reactants, mirroring the procedure described in ref. [38]. In a typical procedure, 0.4328 g of Cd(CH_3_COO)_2_·2H_2_O, 0.1950 g of Na_2_S·9H_2_O, 0.2015 g of Na_2_S_2_O_3_·5H_2_O, and 0.1706 g of C_6_H_8_O_7_·H_2_O are homogenized in a stoichiometric amount and milled.

### 2.3. Characterization Methods and Techniques

X-ray diffraction patterns were obtained on a MiniFlex 600 diffractometer (Rigaku, Tokyo, Japan) in a digital form using copper radiation. Sample analysis modes were as follows: X-ray tube voltage—40 kV, the tube current—15 mA, goniometer movement step size—0.02 2θ, and step time was 0.12 s. During shooting, the sample was rotated in its plane at a speed of 60 rpm. For phase analysis, the ICCD-PDF2 Release 2016 database and the PDXL2 software (Rigaku Corporation, Tokyo, Japan) were used.

The Raman data were obtained using a combined system Solver Spectrum (NT-MDT Spectrum Instruments, Moscow, Russia), equipped with a photomultiplier tube (PMT) detector, a high-stability fast confocal laser (Rayleigh, UK) with imaging and 600/600 grating, and 473 nm solid-state laser excitation. In all experiments, laser power at the sample was 35 mW and the exposure time was 60 s. Continuously tunable filters with ND = 0.5, which reduce the laser intensity by 30%, were also used. When using a blue laser, an error of ±4 cm^−1^ is provided.

The X-ray photoelectron spectroscopy (XPS) measurements were performed in the XPS by Kratos Axis Supra apparatus (Manchester, UK), with a monochromatic Al Kα X-ray radiation, an emission current of 15 mA, and a hybrid lens model. Wide and high-resolution spectra were recorded with a pass energy of 80 eV and 20 eV, using scanning steps of 1 and 0.05 eV, respectively. The obtained data were calibrated by setting the C1s emission at 284.7 eV. The deconvolution and fitting of the interesting elements were carried out using the CasaXPS software (version 2.3.22), by applying a Spine Shirley background in the high-resolution spectra and a Gaussian/Lorentzian line shape for fitting the XPS peaks.

The ultraviolet-visible (UV-Vis) spectra were obtained by a UV-Vis spectrophotometer from Helios Gamma (Thermo Electron Corporation, Rugby, UK) measuring in the range 300–800 nm, using a 1 cm path-length quartz cuvette. The samples were diluted in distilled water by ultrasonic stirring.

The photoluminescence (PL) spectra at room temperature were recorded at the right angle on a photon-counting spectrofluorometer PC1 (ISS, Champaign, IL, USA) with an excitation wavelength of 325 nm. A 300 W xenon lamp was used as the excitation source. Excitation and emission slit widths were set at 1 and 2 mm. A one cm, path-length rectangular quartz cuvette was used. The PL intensity was measured from the powder samples ultrasonically dispersed in absolute ethanol.

Imaging of the samples was performed by transmission electron microscopy (TEM). Prior to TEM analyses, the samples were ultrasonically dispersed in absolute ethanol for a few minutes, then a droplet of the suspension was applied onto a carbon-coated nickel grid. The dried grids were additionally carbon coated to prevent charging of the samples under the high-energy electron beam. TEM analyses were performed using a 200-kV microscope, JEM 2100 (JEOL, Tokyo, Japan) with a LaB6 electron source.

The morphology and size of the products was investigated by a scanning electron microscope, Tescan Vega 3 LMU (TESCAN, Brno, Czech Republic) using an accelerating voltage of 20 kV. In order for the samples to be conductive, the powder was covered by a layer of gold on a FINE COAT ION SPUTTER JFC-1100 fy (JEOL, Akishima, Japan). To obtain the information about chemical composition, the energy-dispersive X-ray spectroscopy (EDS) analyzer, Tescan: Bruker XFlash Detector 410-M (TESCAN, Brno, Czech Republic), was used (in this case, the Au coating was not applied). The same device was used to record elemental maps.

The grain size analysis (DLS) was performed using a particle size laser diffraction analyzer.

Mastersizer 2000E (Malvern Panalytical, Malvern, UK) was used in the dry mode (dry feeder Scirocco 2000M). Each sample was measured three times.

### 2.4. Photocatalytic Measurements

Photocatalytic experiments of CdS samples were performed by recording the UV-visible absorption of the solution after irradiation with a sunlight-simulated lamp, Osram Vita-Lux 300W, with a UV-cutter (λ = 400 nm). The organic dye Orange II was selected as a model solution. The simulated visible light intensity at the surface of the dye solution was 15 mW/cm^2^.

In a typical procedure, 20 mg of a photocatalyst CdS sample was transferred to the 40 mL of Orange II aqueous solution (10 mg/L). Before irradiation, the mixture was stirred for 1 h in a dark regime to attain an adsorption–desorption equilibrium. After that, the lamp was turned on and each 30 min sample was taken for photocatalytic degradation measurements of the sample. Before measurements, the sample was centrifuged to remove the solid CdS photocatalyst from the solution. The filtrate was analyzed with the help of a UV-Vis spectrophotometer SF-56 (LOMO, St. Petersburg, Russia). All photocatalytic tests were carried out in two repetitions to confirm the accuracy of the results obtained.

### 2.5. Hydrogen Evolution

The photocatalytic activity of cadmium sulfide nanoparticles for hydrogen generation experiments was studied using an Osram Ultra-Vita Lux 300 W lamp with a UV cutter (λ ≥ 420) as a source of visible light, as well as an aqueous solution of glycerin (10% by *v*/*v*), which uses glycerin as a sacrificial agent. In the typical photocatalytic experiment, a 30 mg catalyst and 100 mL glycerin solution were placed into a 250 mL tree naked flask and sonicated for 15 min. The photocatalytic reactor was connected to an argon gas flow to remove oxygen molecules, and the dispersed solution was kept in the dark for the first hour. The dispersed solution was magnetic and stirred during throughout experiment. An argon gas flow was fed into the solution through a needle at a rate of 100 mL/min at the degassing (1 h), and 5 mL/min during the photocatalytic experiment. Argon gas served to maintain an inert atmosphere in the reaction system and transport generated hydrogen from the reactor to the chromatograph (Chromos 1000, Dzerzhinsk, Russia).

### 2.6. Antibacterial Activity

Nanoparticles were tested on two bacterial models—*Escherichia coli* ATCC^®^ 25922™ and *Staphylococcus aureus* ATCC^®^ 33591™. The micro-organisms were kept frozen at −20 °C in trypticase soy broth (TSB) medium with 20% *v*/*v* glycerol. Before experiments, micro-organisms were revived on a trypticase soy agar (TSA) medium and incubated at 37 °C.

The nanomaterial was suspended in deionized water, heated at 100 °C for 15 min to ensure sterility, cooled down to room temperature, and sonicated in a water bath sonicator for 30 min. The concentrations used in the analyses were ranging from 6.5 µg/mL to 100 µg/mL, depending on the test.

The antibacterial activity was tested in a 24-h toxicity test, disk assay, and respiration monitoring during the logarithmic growth phase.

For a 24-h toxicity test, bacteria were inoculated at 1:500 from the overnight culture (14–16 h old) to a fresh TSB medium and incubated at 37 °C with agitation (150 rpm). At times of 0 and 24 h, optical density (λ = 600 nm) was measured on a BioTek Synergy H1 (Winooski, VT, USA) spectrophotometer. Afterward, 10% (*v*/*v*) of resazurin (1 mg/mL, Merck, Darmstadt, Germany) was added to the cultures, the samples were incubated at ambient temperature for five minutes, and fluorescence was recorded (λ_ex_ = 520 nm and λ_em_ = 590 nm).

The material for the disk assay was prepared according to EUCAST standards for antimicrobial susceptibility testing [39]. Five µL of nanomaterial (100 µg/mL) was applied to blank disks. Plates were incubated at 37 °C for 18 h and photographed. Four repetitions were prepared for each case.

The cultures for respiration monitoring during the logarithmic growth phase were prepared in the same manner as for the 24-h toxicity test. After 2 or 3 h of incubation (*E. coli* and *S. aureus*, respectively), 10% of resazurin was added. The fluorescence was measured every ten minutes for four hours.

All samples in the 24-h assay were measured in 24 repetitions, whereas each case in the respiration monitoring assay was measured in eight repetitions. The differences between samples were analyzed with One-way ANOVA performed in Origin 2021 software (OriginLab Corp., Northampton, MA, USA) which was also used to visualize the results on figures. Tukey’s posthoc test was used for comparisons. Results with *p* < 0.05 were considered significantly different.

## 3. Results

### 3.1. XRD Results

The X-ray diffraction patterns of CdS obtained from different approaches of mechanosynthesis are shown in Figure 1. The diffraction peaks in the patterns of both cadmium sulfide (aCdS and cCdS) samples match those of cadmium sulfide from the database (PDF2 089-0440). The three main peaks of both samples with 2-theta values of 26.76°, 44.14°, and 52.37° correspond to the three crystal planes of (111), (220), and (311) of cubic phase β-CdS with the mineralogical name, hawleyite. The presence of another polymorph-hexagonal greenockite (PDF2 041-1049) has also been discovered. The broadness of the peaks indicates the nanocrystalline nature of the obtained CdS powders. As shown in the XRD patterns, no other clear peaks were detected, therefore suggesting a high purity of the processed compounds.

### 3.2. Raman Spectroscopy Results

Raman spectra of mechanochemically prepared CdS nanoparticles are presented in Figure 2. All characteristic peaks for CdS are almost the same for both samples. Usually, the Raman spectrum of the CdS exhibits a well-resolved band at 302 cm^−1^, corresponding to the first-order scattering of the longitudinal optical (LO) phonon mode and the second-order band at approximately 599 cm^−1^. Also, CdS can have both hexagonal wurtzite and cubic zinc blended structures. The zone-center, longitudinal–optical A1 (LO) phonon frequency for both structures are nearly 305 cm^−1^ [40]. In this paper, the Raman spectrum of cadmium sulfide prepared by mechanochemical method contains three distinguishable peaks at 300.40, 599.44, and 910.42 cm^−1^ (acetate route (aCdS)), and 300.40, 592.42, and 900.42 cm^−1^ (combined approach (cCdS)), respectively. These values are in accordance with the main wavenumber peaks of CdS [41]. As a blue laser during shooting reveals an error with a ±4 cm^−1^ wavenumber, a negligible difference in LO2 and LO3 wavenumbers in the Raman pattern can be ignored. Detailed characterizations of CdS samples by Raman spectroscopy can be found in [42].

### 3.3. X-ray Photoelectron Spectroscopy Results

The results of XPS analysis are presented in Figure 3. The XPS measurements on the processed aCdS and cCdS compounds revealed the presence of traces from the reactants, such as sodium, and carbon or oxygen from organic compounds that could get absorbed in the surface of the fine powders during their processing. However, those impurities were very low in amount, e.g., on average less than 0.3 % in the case of Na. Besides the extra residues, the XPS spectra allowed determination of the surface chemical composition of the processed CdS compounds by analyzing the characteristic core-level emissions of sulfur and cadmium.

For the Cd, the 3d core-level spectra decomposition showed the presence of the characteristic doublet with the 3d5/2 at ~405.5 eV, and additional doublets were found at a lower binding energy (~405.2 eV), which are both related to Cd^2+^ species [43]. A load of sulfur connected to the CdS particles was evidenced by the 2p spectra, which were decomposed in two peaks positioned at 161.6 and 162.7 eV, which are connected to the S 2p3/2 and S 2p1/2 spin-orbit levels of sulfur in the CdS, respectively [44]. Additional peaks at a higher binding energy between 165–170 eV revealed a grade of oxidation in the CdS compounds. The doublets positioned at 166.8, 168.3, and 168.9 eV are connected to sulfate species formed by the oxidation of the sulfur-based compounds in air or during high-energy milling. The aCdS sample seems to be more oxidized. However, these peaks have been attributed previously to oxidized sulfur species in the form of SO_x_^2−^ (x = 3 or 4), which are located on the CdS particle’s surface [45].

### 3.4. UV-Vis Spectroscopy Results

The optical properties of aCdS and cCdS samples were investigated by ultraviolet-visible (UV-Vis) absorption spectroscopy (Figure 4 and Figure 5). The optical bandgap energy of prepared samples was calculated from the empirical formula (Equation (2))
(2)$${\left(\mathrm{ah}\nu \right)}^{n}=A\;\left(h\nu -Eg\right)$$
where *E_g_* is the optical bandgap energy, h is Planck constant, a is the absorption coefficient at ν frequency, A is constant, and *n* = 2, 1/2, and 3/2, correspondingly.

UV-Vis absorption spectra of aCdS and cCdS samples (Figure 4) exhibit a broad absorption in the whole ultraviolet-visible region with the absorption maximum in the visible region. Figure 4 shows UV-Vis spectra with a clearly higher intensity of light absorbance evidenced in the case of the aCdS sample, in comparison with cCdS.

The optical band gap, *E_g_*, was determined based on the Tauc Equation (2) for directly allowed optical transition, αhν = A (hν − *E_g_*)^1/2^, by the plotting (αhν)^2^ against photon hν energy and extrapolating the slope in the band edge region to zero, as shown in Figure 5. The optical bandgap was determined to be 2.36 eV for aCdS and 2.78 eV for cCdS, respectively. The optical bandgap of aCdS sample is red-shifted and cCdS is blue-shifted in comparison with the bulk CdS, which has a characteristic bandgap energy of about 2.4 eV [46]. The determined bandgap for cCdS (2.78 eV) is in accordance with the results published in previous papers [47]. This blue shift indicates the presence of quantum-size effects in the cCdS sample. However, the determined bandgap for aCdS (2.36 eV) is in agreement with other reports [48,49]. The slightly red shift could originate from the surface defects of the CdS nanoparticles [50]. These shifts can be due to the influence of various factors such as grain size, structural parameters, carrier concentration, presence of impurities, deviation from stoichiometry, lattice strain, etc.

### 3.5. PL Spectroscopy Results

The optical properties of aCdS and cCdS samples were also analyzed by photoluminescence (PL) emission spectroscopy (Figure 6). The room-temperature PL spectra of aCdS and cCdS samples were taken using an excitation wavelength of 325 nm (3.8 eV) and are displayed in Figure 6.

The PL spectra of aCdS and cCdS samples are a little different, but both show emission spectra in the visible range. The aCdS sample has broad emission bands at 550 nm (2.2 eV) and 580 nm (2.1 eV), almost consistent with the literature [51]. However, the cCdS sample shows the broad emission bands at 470 nm (2.6 eV) and 490 nm (2.5 eV), which are in accordance with results reported in previous papers [52]. CdS nanoparticles show two types of PL emissions, identified as band-edge (400–500 nm) and surface defect, or surface state emissions (500–700). In our case, the observed green emission at 550 nm, for sample aCdS, can be assigned to charge the carrier recombination at the surface states of CdS nanoparticles. This recombination was reported as a radiative recombination of the free charge carrier and trapped charge carriers at surface defects [53]. The yellow emission observed at 580 nm (2.1 eV) for sample aCdS is less than the bandgap energy calculated (2.21 eV); the luminescence may be due to transitions involving defect states. The source of defect states may be associated with sulfur vacancies, intrinsic defects, or impurities [54]. In the case of the cCdS sample, the blue emission at around 470 nm (2.6 eV) is attributed to direct recombination of electron and hole pairs at the bandgap that cause the band edge luminescence [55]. The results for synthesized CdS nanoparticles in contrast to bulk CdS indicate that the emission depends on the particle size, shape, preparation method, etc.

### 3.6. SEM Results

The morphology of the prepared aCdS and cCdS samples was investigated by SEM (Figure 7). The sample prepared by a traditional acetate route contains finer particles, while in the case of cCdS, the fine particles are embedded into the matrix of large grains the size in tens of microns. It seems that the presence of an additional source of sulfur that leads to the formation of needle-like structures of CuS [38] results in the formation of large agglomerates in the present case. The distribution of Cd and S elements in both samples is homogeneous, and the locations perfectly match each other (Appendix A), thus providing another proof of successful CdS synthesis. The EDS analysis (Appendix A) has shown almost ideal stoichiometry for the CdS products, albeit the content of Cd is slightly higher in all cases. The small sulfur deficiency is common for the mechanochemical synthesis of sulfides [56].

### 3.7. TEM Results

Figure 8 shows the results of TEM analyses of the two samples. Low-magnification images of the samples with selected area diffraction (SAED) patterns shown in Figure 8a,d reveal that the samples are composed of agglomerated nanoparticles, which is a typical product morphology in mechanochemical synthesis. The d-values of the diffraction rings in both samples are in agreement with the results of XRD analyses, and indicate that the sphalerite-type (cubic) modification is the prevailing one in both samples. There are two main differences between the samples. According to the diffuse diffraction rings of the aCdS sample, it is composed of very small nanoparticles with an average size below 10 nm; whereas the SAED pattern of the cCdS sample contains reflections dots in addition to the diffuse rings, indicating the presence of slightly larger nanoparticles in addition to the smaller ones. The second difference between the samples is the presence of a fairly high fraction of amorphous material in addition to the crystalline CdS nanoparticles in the cCdS sample. High-resolution TEM analysis of the aCdS sample confirmed that it is composed of very small crystallites with a diameter of 10 nm or lower (Figure 8b). A more detailed analysis revealed that the grains are composed of sphalerite-type slabs that contain wurtzite-type defects, or even layers with the wurtzite-type structures (Figure 8c). We estimate that the sphalerite modification prevails in the grains of both samples; whereas the wurtzite-type sacking is mainly present at defects and thinner sections formed as a consequence of high-energy milling. The presence of both structural types is possibly due to the close structural relationship of the two modifications, as shown in the structural models in Figure 8c. A single layer is identical in both structures, but the stacking of layers is different. In sphalerite type, the subsequent layers (the (111)_Sp-t_ layers) have an identical orientation, which results in the cubic close-packed stacking (ccp: -A-B-C-) of the anionic sublattice; whereas in the wurtzite-type modification, the subsequent layers (here, these are the (001)_Wur-t_ layers) can be described by in-plane rotation of 180°, and this results in hexagonal close-packed stacking (hcp: -A-B-A-B-). Due to this close structural relationship between both structural types, intergrowths are common in CdS and similar compounds, e.g., ZnS, and can form for different reasons. For example, these happen as a result of crystallization near the Sph–Wur transition temperature or mechanically induced crystal slip due to high-energy milling or as observed in CdS samples synthesized in this work.

### 3.8. DLS Analysis Results

Results of DLS analysis of mechanochemically synthesized aCdS and cCdS samples are presented in Figure 9. The grains of cCdS sample are significantly coarser, as the maximum grain size is reached around 107 µm and the overall size distribution is polymodal. On the other hand, a bimodal grain size distribution has been observed in the case of aCdS, prepared using the traditional acetate route, with the maximum being located slightly below 3 µm and being detected for the smaller fraction. A significantly minor, coarser fraction for this sample has a maximum located around 134 µm. The coarser character of the cCdS sample is in accordance with the observation of SEM analysis (Figure 7).

### 3.9. Photocatalysis Results

Comparison of the photocatalytic activity of mechanochemically synthesized aCdS and cCdS (Figure 10) revealed that both substances can degrade almost all molecules of Orange II after 180 min of visible light irradiation. In general, uniform photodegradation of the model solution without sharp fluctuations is observed. However, cCdS was a more active photocatalyst than aCdS.

The study of the kinetics of the photocatalytic degradation of Orange II is shown in Figure 11. The rate of the photocatalytic reaction was accepted as a pseudo-first-order reaction and described with Langmuir–Hinshelwood kinetics [57]:(3)$$\ln\left(\frac{C_{0}}{C}\right)=\mathrm{kt},$$
where C_0_ and C are initial and final concentrations of Orange II solution in the moment of time, t, and k (min^−1^) is the rate constant of the reaction. The correlation coefficient, *R^2^*, served as proof of the kinetic order of organic dye degradation. The values of the k and *R^2^* are presented in Table 1. The rate constant (Table 1) of photocatalytic decomposition of Orange II by cCdS catalyst (k = 0.018 min^−1^) is 1.5 times higher than the aCdS catalyst (k *=* 0.012 min^−1^). This difference explains the effect of the shape of nanoparticles on the ability to rapidly excite an electron in a semiconductor and slow down the rate of recombination of an electron and an electron-hole, and thereby the effect on the photocatalytic activities of cadmium sulfide nanoparticles. Similarly, the higher photocatalytic activity can be compromised by the surface chemistry of the particles, and as revealed by the XPS spectra, the presence of oxidized species in the aCdS compounds was higher than in the cCdS, which increases their electrical resistivity [58]. cCdS nanoparticles have a greater propensity for photocatalytic activity on the degradation of organic compounds than usually spherical aCdS nanoparticles.

Appendix A reveals the results from the cyclic test of aCdS and cCdS samples for Orange II degradation. The degradation activity of both samples for Orange II exhibits only a slight decrease after five circles. This proves that both samples have a good cyclic utilization. After checking the stability of photocatalysts, utilized aCdS and cCdS samples were centrifuged at 4000 rpm for 10 min and dried for 12 h to perform XRD analysis.

Appendix A shows the XRD patterns of both CdS photocatalysts after performing cyclic tests. The diffraction peaks of CdS in the patterns of the samples closely match those of cadmium sulfide from the database (JCPDS 89-0444). In addition to the peaks of cadmium sulfide, both samples contain cadmium carbonate (CdCO_3_; ICCD PDF card No. 00-001-0907), which clearly shows that some Cd(II) ions are leached out from the process. The presence of carbonate ions might come from the interaction between the solution with the photocatalayst and ambient atmosphere.

In comparison with others, photocatalysts such as TiO_2_ and ZnO, and the CdS nanoparticles have a significantly narrow band gap energy and demonstrate that they can absorb a broad absorption sunlight spectrum. Moreover, the photogenerated e^−^ and h^+^, due to the high surface-to-volume ratio, can access the particle surface more efficiently and can be easily captured by redox couples in the solution with less recombination [59].

Spectral analysis results show that the band gap of the cCdS nanoparticles was 2.78 eV, which is larger than the band gap of bulk CdS (2.42 eV) and acetate-method-synthesized aCdS nanoparticles (2.36 eV). When the visible light has a wavelength less than 448 nm, with an energy of hn that matches the energy of the band gap (Eg) of the cCdS nanoparticles, it generates electron (e^−^) and hole (h^+^) pairs with strong oxidizing and reducing properties:(4)$$2\mathrm{CdS}+h\nu \;\to \mathrm{CdS}\;\left(e^{-}\right)+\mathrm{CdS}\;\left(h^{+}\right)$$

The large band gap of cCdS nanoparticles forms to a nonradiative recombination of e^−^ and h^+^ pairs [60], which enhances photocatalytic activity. Water (H_2_O) adsorbed on the surface of CdS nanoparticles captures the hole (h^+^) and is oxidized to form a hydroxyl radical (^•^OH), which is expressed as:(5)$$H_{2}O+\mathrm{CdS}\;\left(h^{+}\right)\to {\;\mathrm{OH}}^{•}+{\;H}^{+}+\mathrm{CdS}$$

Also, the presence of oxygen (O_2_) prevents the recombination of pairs of electrons (e^−^) and holes (h^+^), and it leads a radical anion (^•^O_2_^−^), accepting electrons (e^−^) from the conduction band and further combining with a proton to give ^•^OOH:(6)O2+CdSe− → O2−•+CdSO2−•+H+ → OOH•

Formed radicals, ^•^OH and ^•^O^2−^, decompose organic compound Orange II during the photocatalytic process by reacting to the aromatic ring of Orange II molecules and opening it at the azo bond and the hydroxylated ring, finally giving gaseous N_2_, H_2_O, and CO_2_ [61]:(7)Orange II+OH•/O2−• → products N2, H2O and CO2

In Table 2, results of photocatalytic tests of CdS nanoparticles prepared with different approaches for recent years are reviewed. According to the collected data, the photocatalytic activities of our samples are in no way inferior to those obtained by other methods. Since our approach is environmentally friendly, as no harmful organic solvents are used and only 5 min of the treatment at room temperature was necessary to obtain CdS nanoparticles, it is more beneficial than the ones reported in the Table 2.

### 3.10. Hydrogen Evolution Results

Figure 12 shows the results of a photocatalytic experiment on hydrogen generation using photocatalysts aCdS and cCdS, where the maximum values of hydrogen formation are recorded in the first hour of photocatalysis for aCdS and the second hour of photocatalysis for cCdS, and they are equal to 4.1 and 7.8 μmol g^−1^ h^−1^, respectively. After 2 h of photocatalysis, the activity of cadmium sulfide photocatalysts drops significantly, which indicates their unstable structure. The results of photocatalytic experiments on hydrogen generation showed that cadmium sulfide obtained by combined mechanosynthesis (cCdS), once again, confirms its superiority over cadmium sulfide synthesized by the traditional acetate method (aCdS). The manifestation of a higher photocatalytic activity during hydrogen generation of cCdS nanoparticles compared to aCdS nanoparticles can be associated with the size of crystallites, morphology, and structural defects of crystal lattices of cadmium sulfide nanoparticles. From the XRD results, the diffraction peaks of cCdS nanoparticles are slightly wider, and this cCdS average crystallite size is slightly smaller than those of aCdS. A decrease in the crystallite size improves the photocatalytic activity of nanoparticles. From the results of TEM, it can be seen that there is greater content of crystal defects in cCdS nanoparticles, which should also be the reason for the higher photocatalytic activity of cCdS. The large band gap of cCdS nanoparticles (*Eg* = 2.78 eV) is also increased with hydrogen generation performance, due to strong oxidizing and reducing properties [59]. A decrease in hydrogen generation on the surface of both CdS nanoparticles is observed after 120 min, but the performance of cCdS remains higher than that of aCdS. The decrease in hydrogen generation could be caused by the increase in the concentration of glycerol oxidation products during photocatalysis in the solution [69]. Glycerol was used as a sacrificial agent for generating hydrogen.

### 3.11. Antimicrobial Activity

To demonstrate the multidisciplinary application of the mechanochemically prepared CdS nanoparticles, their antibacterial potential was also investigated. The measurements of optical density shown in Figure 13 did not show any significant differences between the two samples in the case of *S. aureus*. On the other hand, in this test, all samples of *E. coli* had significantly higher optical density than the control sample (Appendix A). This observation confirms previous findings that bacteria could build their population after a long incubation, even if it is initially inhibited [70]. This was also confirmed in the disk diffusion test that the current study did not produce any halo zones around the nanomaterial, suggesting that ions were not being released to the agar medium (Appendix A) [71]. The respiration recorded in *E. coli* was not significantly different from the control sample (Figure 14), whereas the samples containing the highest concentrations of nanomaterials showed significantly higher respiration in *S. aureus* (Appendix A). This phenomenon could be explained by the inhibition observed during the respiration monitoring in the logarithmic phase (Figure 15). This experiment has clearly shown that nanomaterials could actively inhibit the growth of both micro-organisms, starting from the smallest concentration. Judging from the curves obtained within this step, the application of aCdS nanoparticles resulted in higher growth inhibition than in the case of cCdS. The probable cause for this effect was the difference in grain size distribution recorded for these nanoparticles. The data confirmed the concept that a nanomaterial’s toxicity increases with the decrease in its size [72]. The inhibition was increasing with the concentration of the used nanomaterial. In this view, higher respiration of *S. aureus* after 24-h incubation may be a result of the retardation that occurred during the logarithmic growth. Staphylococci have a longer generation time than *E. coli.* Therefore, the effect could still be observed in this case [73,74]. Even though the inhibition was not complete in the cultures, the results have shown that the produced nanomaterials can be considered antibacterial and included in a broader group of antimicrobial agents that are used for various applications [75,76].

## 4. Conclusions

In summary, CdS nanoparticles have been successfully obtained by a mechanochemical pathway, either using the acetate route (aCdS) or its combination with sodium thiosulfate, an additional source of sulfur (cCdS). The results of UV-Vis spectroscopy revealed that the bandgap energy of cadmium sulfide (2.78 eV), produced via combined milling (cCdS), is higher than the aCdS sample prepared by acetate route (2.36 eV). The rate constant of photocatalytic degradation of dye by cCdS was 1.5 times higher than aCdS. The antibacterial properties of both CdS samples were tested on reference strains of *E. coli* and *S. aureus*. Results of experiments have confirmed that the CdS samples have some capability to inhibit bacterial growth during the logarithmic growth phase, in which the effect of aCdS was more potent than in the case of cCdS.

## Figures and Tables

**Figure 1 nanomaterials-12-01250-f001:**
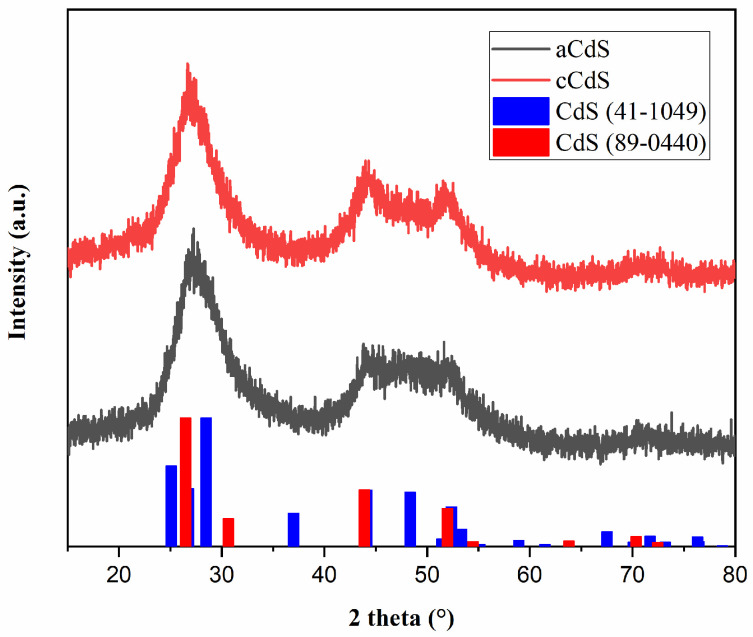
XRD patterns of mechanochemically synthesized aCdS and cCdS samples.

**Figure 2 nanomaterials-12-01250-f002:**
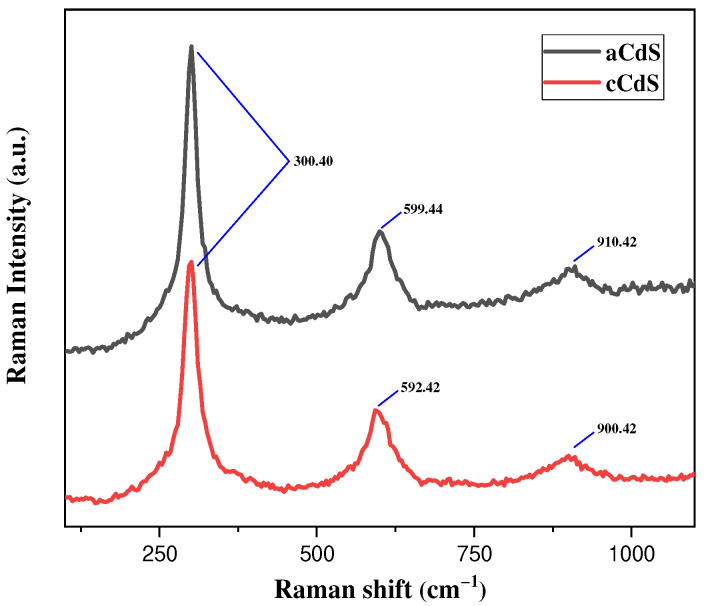
Raman spectra of as-synthesized aCdS and cCdS samples.

**Figure 3 nanomaterials-12-01250-f003:**
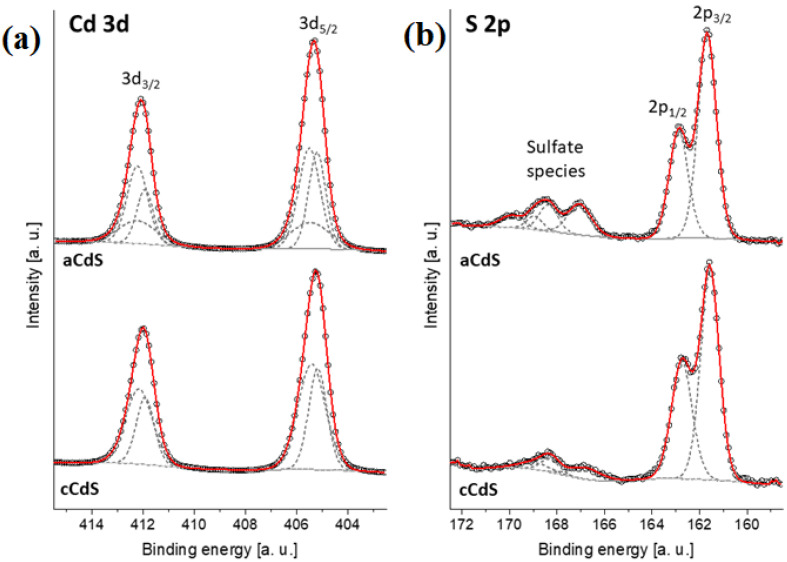
Core level emissions of (**a**) Cd 3d and (**b**) S 2p from the XPS spectra of mechanochemically synthesized aCdS and cCdS powders.

**Figure 4 nanomaterials-12-01250-f004:**
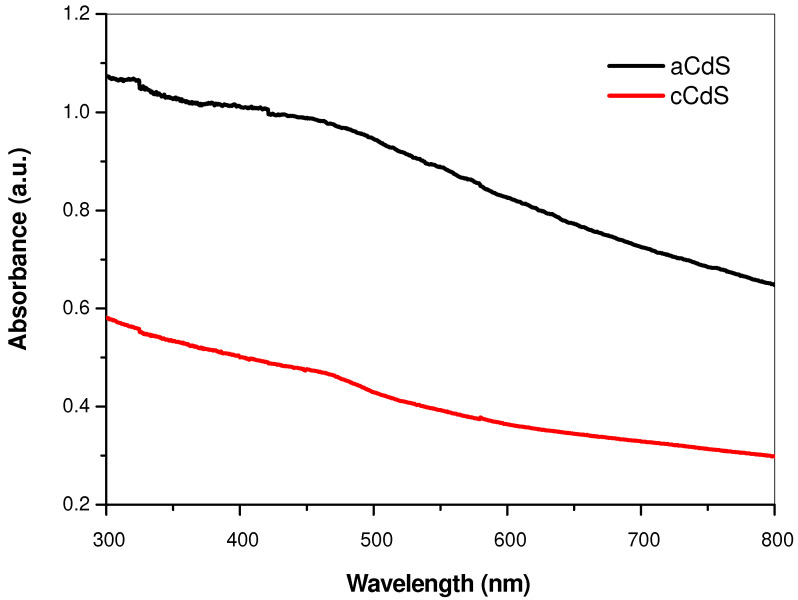
UV-Vis absorption spectra of aCdS and cCdS samples.

**Figure 5 nanomaterials-12-01250-f005:**
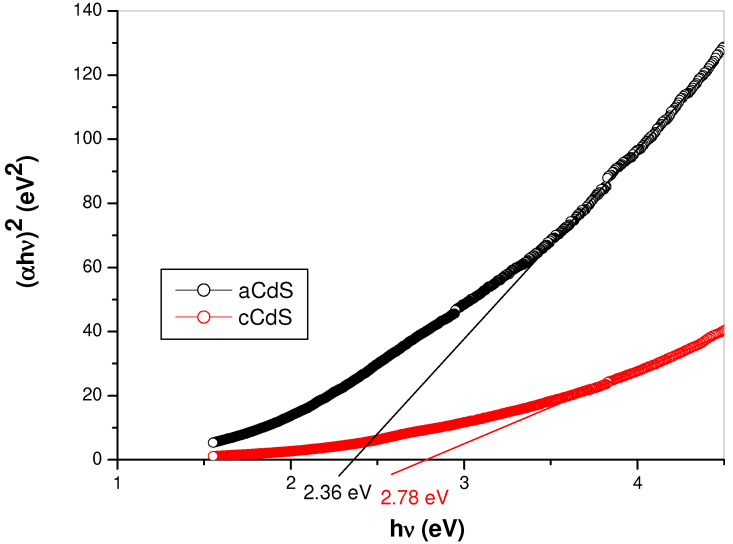
Tauc’s plots with the determined bandgap energy of aCdS and cCdS samples.

**Figure 6 nanomaterials-12-01250-f006:**
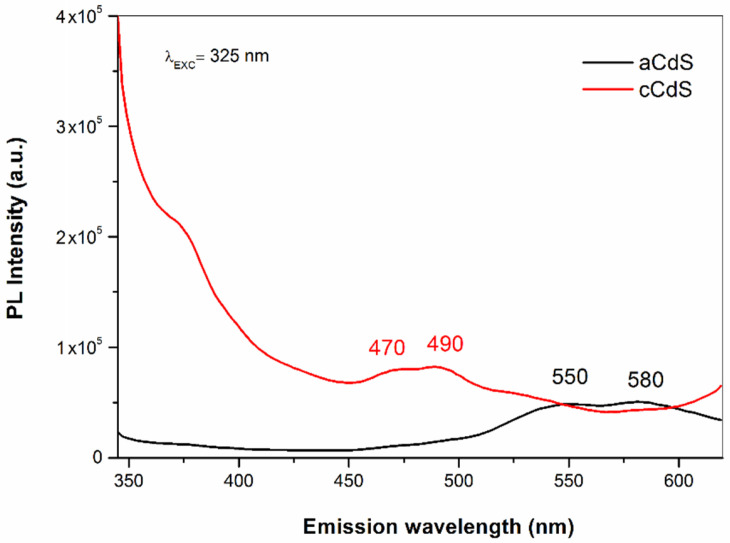
PL emission spectra of aCdS and cCdS samples.

**Figure 7 nanomaterials-12-01250-f007:**
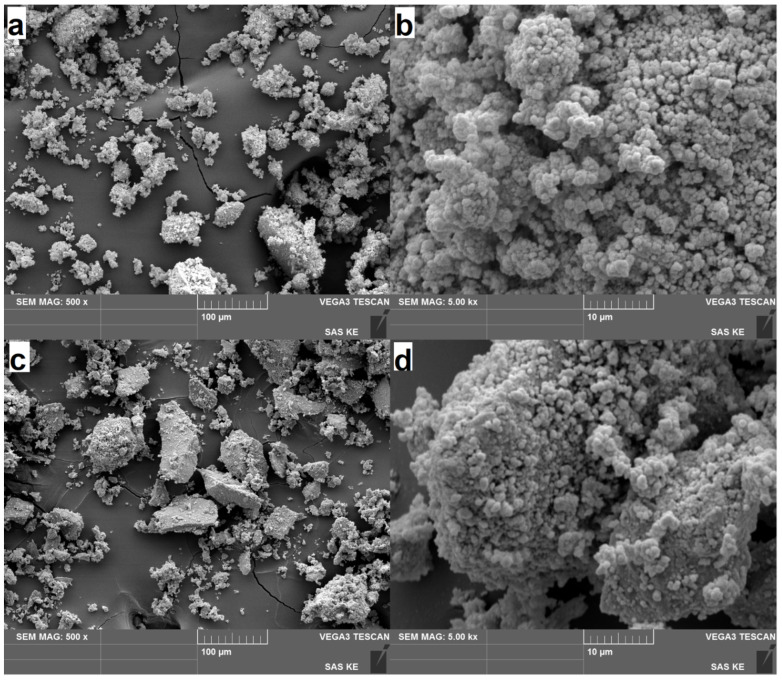
SEM images of the mechanochemically synthesized CdS samples: (**a**,**b**) aCdS; (**c**,**d**) cCdS.

**Figure 8 nanomaterials-12-01250-f008:**
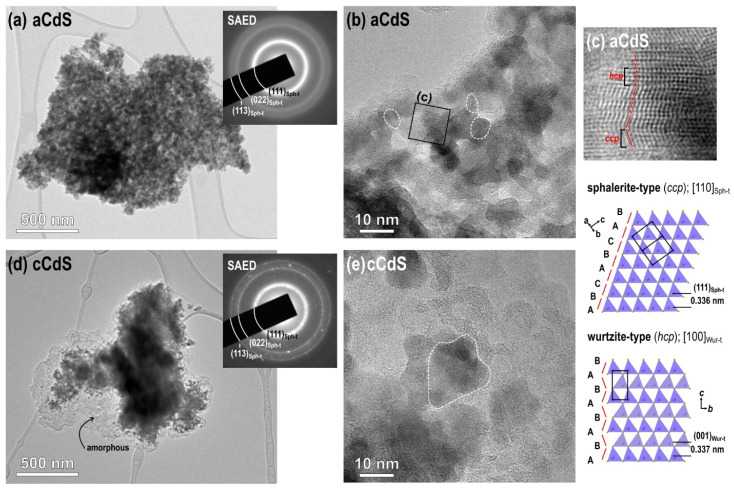
TEM analysis of the mechanochemically synthesized CdS samples: low-magnification images of the (**a**) aCdS and (**d**) cCdS samples with SAED; high-resolution TEM analysis of the (**b**) aCdS and (**e**) cCdS samples; (**c**) detailed analysis of aCdS sample.

**Figure 9 nanomaterials-12-01250-f009:**
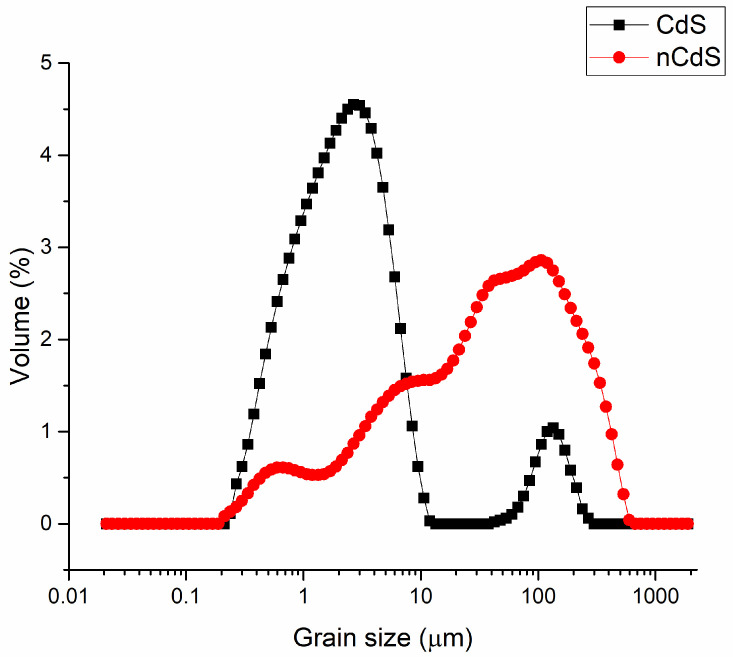
Grain size distribution of aCdS and cCdS samples.

**Figure 10 nanomaterials-12-01250-f010:**
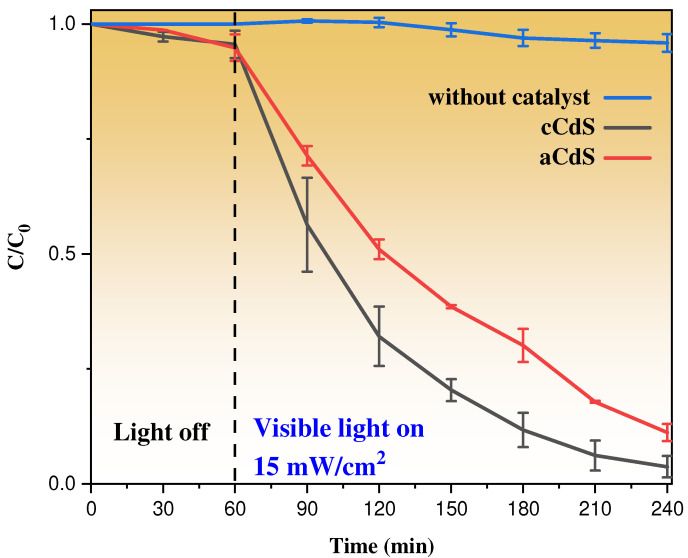
Photocatalytic degradation profiles of mechanochemically synthesized aCdS and cCdS catalysts under visible light irradiation.

**Figure 11 nanomaterials-12-01250-f011:**
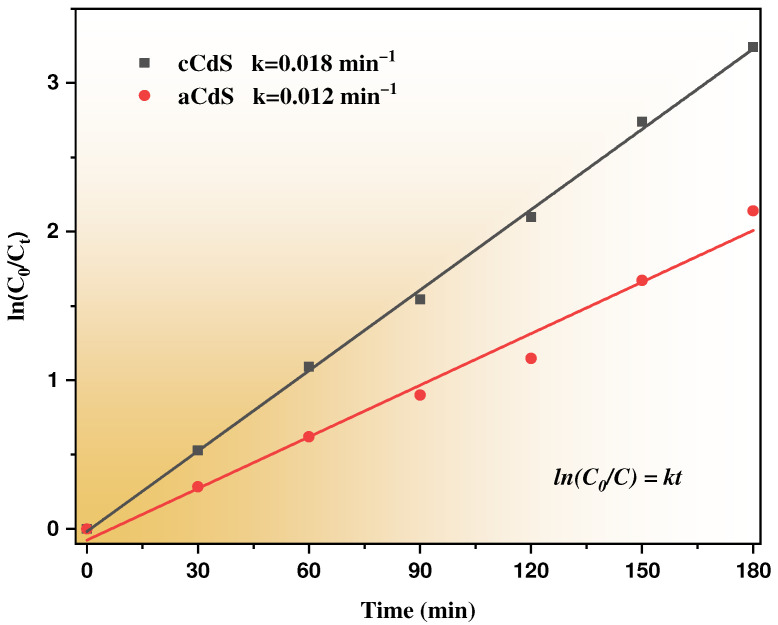
Kinetic linear simulation curves for Orange II photocatalytic degradation over aCdS and cCdS samples.

**Figure 12 nanomaterials-12-01250-f012:**
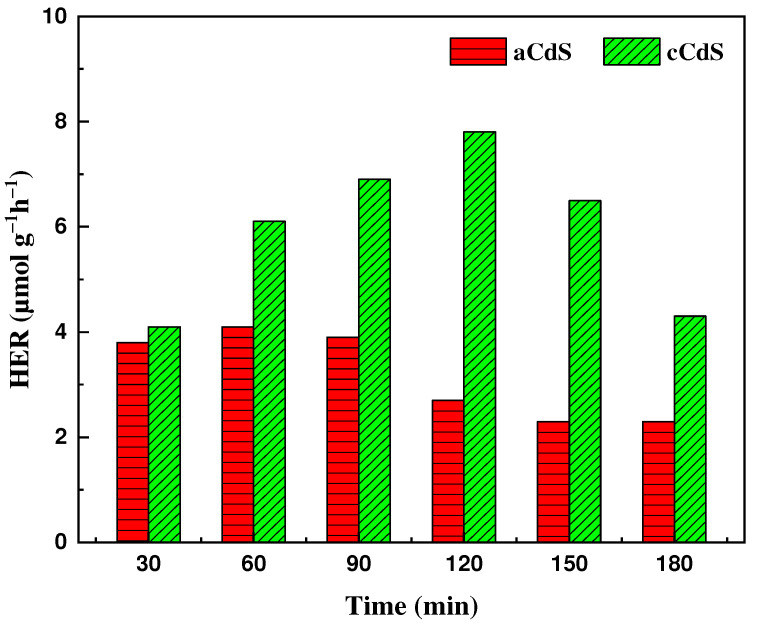
Photocatalytic H_2_ production over aCdS and cCdS samples.

**Figure 13 nanomaterials-12-01250-f013:**
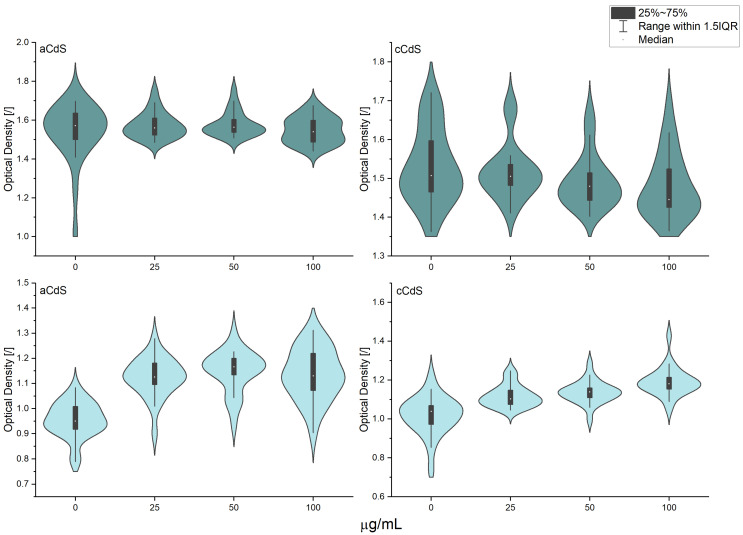
Optical density of 24-h cultures of *S. aureus* (first row) and *E. coli* (second row) contacted with CdS nanoparticles or deionized water.

**Figure 14 nanomaterials-12-01250-f014:**
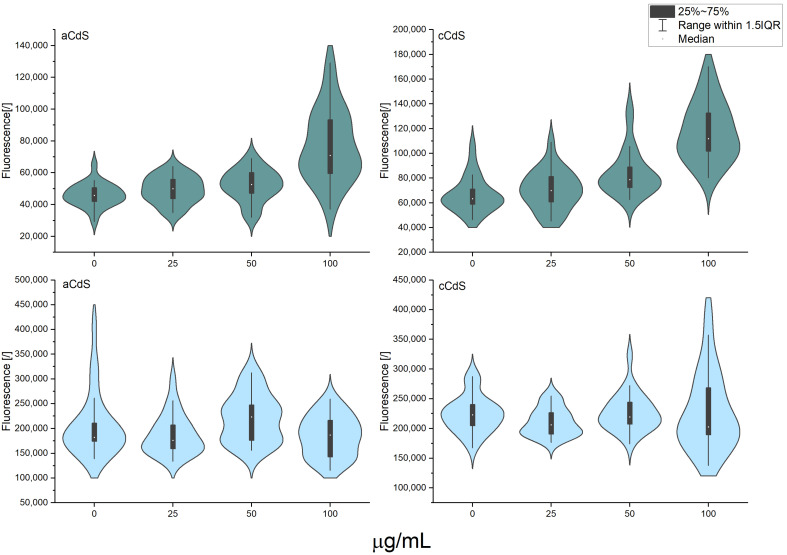
Respiration of cultures after 24-h incubation with nanomaterials or deionized water; *S. aureus* (first row) and *E. coli* (second row).

**Figure 15 nanomaterials-12-01250-f015:**
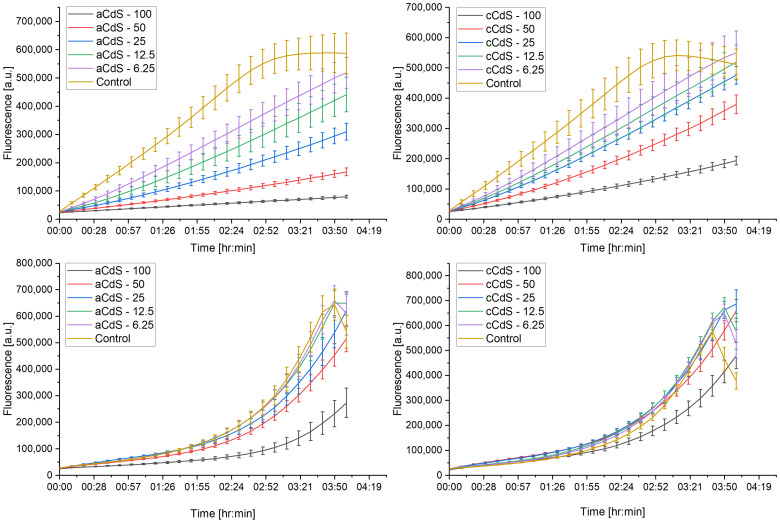
Respiration of cultures in the logarithmic growth phase in resazurin assay; *S. aureus* (first row) and *E. coli* (second row).

**Table 1 nanomaterials-12-01250-t001:** The parameters for the pseudo-first-order photocatalytic reaction of aCdS and cCdS.

The Sample	k, min^−1^	*R^2^*
aCdS	0.012	0.98
cCdS	0.018	0.98

**Table 2 nanomaterials-12-01250-t002:** Comparison of photocatalytic activities of CdS nanoparticles with other methods.

№	Synthetic Method	Experimental Conditions		Precursors	Degraded Dye, Concentration	Photocatalytic Efficiency	Rate Constant (min^−1^)	[Ref]
		Time	Temperature (°C)					
1	Composite-molten-salt (CMS)	24–72 h	160–220	Cd(NO_3_)_2_·4H_2_O, Na_2_S·9H_2_O, LiNO_3_, KNO_3_	MB, 4 mg/L	76.3%@140 min	-	[62]
					RhB, 8 mg/L	94.9%@140 min	-	
2	Solvothermal	6 h	180	CdCl_2_·5H_2_O, CS(NH_2_)_2_	MB, 6 mg/L	95%@80 min	0.0365	[63]
3	One-step solid-state reaction	30 min	-	Cd(CH_3_COO)_2_·2H_2_O, Na_2_S_2_O_3_·5H_2_O	RhB, 10 mg/L	95%@80 min	0.0429	[64]
4	Biogenic synthesis	72 h	28	Strain of T. Harzianum, CdCl_2_, Na_2_S	MB, 10 mg/L	37.15%@60 min	0.0076	[59]
5	Hydrothermal	24 h	-	Cd(Ac)_2_·2H_2_O, PVP-K30, CS(NH_2_)_2_	MO, 20 mg/L	93.3%@240 min	-	[65]
6	Sonochemical	1 h	RT	Cd(CH_3_COO)_2_, Na_2_S, tryptophan	MO, 5 × 10^−6^ M	75.33%@240 min	0.0062	[66]
7	Photochemical	24 h	-	CdSO_4_, Na_2_S_2_O_3_	MO, 8 × 10^−6^ M	26.3%@70 min	0.0058	[67]
8	Commercial CdS	-	-	-	MO, 10 mg/L	78%@90 min	-	[68]
9	Mechanochemical	5 min	RT	Cd(CH_3_COO)_2_·2H_2_O, Na_2_S·9H_2_O, Na_2_S_2_O_3_·5H_2_O, C_6_H_8_O_7_	Orange II, 10 mg/L	93%@180 min	0.018	this work

## Data Availability

Not applicable.

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
