# Peer review of "Sustainable Synthesis of Cadmium Sulfide, with Applicability in Photocatalysis, Hydrogen Production, and as an Antibacterial Agent, Using Two Mechanochemical Protocols"

_nanomaterials, 2022, doi:10.3390/nano12081250_

Round 1

Reviewer 1 Report

The work presented in the manuscript nanomaterials-1622935 reports the synthesis of CdS nanoparticles via two different method and then explored the significant features of the obtained CdS. Among the two methods, combined mechanochemcial route enabled the formation of CdS NPs with good photocatalytic activity. Overall the manuscript is interesting and suitable for publication after the following revisions. 

  1. Authors should discuss the Raman analysis of CdS samples in detailed. The analysis can be found in the following published paper and cite at appropriate place.  Nanomaterials 202010(4), 619; https://doi.org/10.3390/nano10040619
  2. The provided UV-vis absorption spectra is not so good. It is advised to provide UV-vis diffuse reflectance spectra in powder form. If not possible, then record in water media. In addition, it is advised to provide the physical images of obtained samples.
  3. Authors should give provide the details how the cCdS shown higher H2 evolution activity and what is the reason for decrease of activity after 120 min.

Author Response

Reviewer’s comments Answers for comments
Authors should discuss the Raman analysis of CdS samples in detailed. The analysis can be found in the following published paper and cite at appropriate place. Nanomaterials 2020, 10(4), 619; https://doi.org/10.3390/nano10040619 The results of Raman analysis are now discussed in detail. We have also cited the reference proposed by the reviewer. 
The provided UV-vis absorption spectra is not so good. It is advised to provide UV-vis diffuse reflectance spectra in powder form. If not possible, then record in water media. In addition, it is advised to provide the physical images of obtained samples. Our experimental setup did not allow us to measure UV-Vis DRS spectra, therefore we recorded the UV-Vis spectra in water according to this reviewer’s suggestion. The measured spectra in water were slightly better than one provided in ethanol. The newly recorded spectra are now inserted in the revised manuscript, replacing the ones measured in ethanol in the original submission.
Authors should give provide the details how the cCdS shown higher H2 evolution activity and what is the reason for decrease of activity after 120 min. The manifestation of a higher photocatalytic activity during hydrogen generation of cCdS nanoparticles compared to aCdS nanoparticles can be associated with the size of crystallites, morphology, and structural defects of crystal lattices of cadmium sulfide nanoparticles. From the XRD results, the diffraction peaks of cCdS nanoparticles are slightly wider, and this cCdS average crystallite size is slightly smaller than those of aCdS, and a decrease in the crystallite size improves the photocatalytic activity of nanoparticles. From the results of TEM, it can be seen a greater content of crystal defects in cCdS nanoparticles, which should also be the reason for the higher photocatalytic activity of cCdS. The large band gap of cCdS nanoparticles (Eg=2.78 eV) is also increase hydrogen generation performance due strong oxidizing and reducing properties. A decrease in hydrogen generation on the surface of both CdS nanoparticles is observed after 120 minutes, but the performance of cCdS remains higher than that of aCdS, the decrease in hydrogen generation could be caused by the increase in the concentration of glycerol oxidation products during photocatalysis in solution. Glycerol was used as a sacrificial agent for generating hydrogen.

Reviewer 2 Report

In this paper, different CdS particles were prepared by mechanochemical methods and their performance in photocatalytic hydrogen evolution and bacterial inhibition were presented. However, the novelty of the current system is insufficient. Meanwhile, here are some points of weakness:

  • Lack of comparison with the CdS or commercial CdS nanoparticles prepared by other methods;
  • The characterization results are just presented without logic behind them;
  • The stability of the current systems are not satisfied;
  • There are too many references.

Author Response

Reviewer’s comments Answers for comments
Lack of comparison with the CdS or commercial CdS nanoparticles prepared by other methods; The comparison with the commercial CdS nanoparticles prepared by other methods was not included in the scope of our study. The purpose of our study was to compare the photocatalytic activity of mechanochemically prepared cCdS and aCdS.
The characterization results are just presented without logic behind them; From our point of view, the characterization results are presented in a logical order. Firstly, we presented synthesis results and then application results. The manuscript was structured according to the template.
The stability of the current systems are not satisfied; From our experience, mechanochemically prepared CdS is stable, and it is not oxidizing very quickly. Because it is a powder, the stability should be higher than in the case of nanosuspension. We could not figure out what exact measurements to determine the stability the reviewer had in mind. If he can be more specific, we can perform some additional experiments.
There are too many references. We agree with this Reviewer’s comment. Some number of references was removed from manuscript to decrease their amount.

Reviewer 3 Report

The authors have fabricated CdS NPs by two routes including acetate route (aCdS) and combined mechanochemical synthesis (cCdS). The materials were tested for the oxidation of orange II, H2 evolution and as antibacterial agent. My comments are listed below:    

  • Figure 14. is not readable. I suggest to remake it.
  • In terms of photocatalytic oxidation of Orange II, I suggest to add the results of photolysis to Figure 9 for the purpose of comparison.
  • Since the Cd is very toxic, did the authors check the stability of the photocatalyst? The leaching of Cd from the photocatalyst solid may lead to toxicity issue.  
  • I think there is an issue regarding the explanation of results. In the conclusion section, the authors stated that cCdS was more effective than aCdS because cCdS has a band gap higher than aCdS. However, this statement is not convincing. Usually, a lower band gap allows better light absorption and a higher yields of generated redox charges. I suggest to re-consider this statement.   

Author Response

Reviewer’s comments Answers for comments
Figure 14. is not readable. I suggest to remake it. Figure 14 is updated and renamed as “Figure 15”.
In terms of photocatalytic oxidation of Orange II, I suggest to add the results of photolysis to Figure 9 for the purpose of comparison. Photocatalytic degradation of Orange II without catalyst under visible light irradiation was added to Figure 9. Almost no degradation was observed.
Since the Cd is very toxic, did the authors check the stability of the photocatalyst? The leaching of Cd from the photocatalyst solid may lead to toxicity issue.   The reviewer is right that soluble form of Cd is toxic, however, due to low solubility of CdS, this form shows much lower toxicity. The stability test of both CdS samples for Orange II dye was performed. Results of cyclic test are presented in ESI (Figure S5). For determination of toxicity Cd, solid residual after the cyclic tests were centrifuged, dried, and analyzed with XRD to the presence of Cd in utilized photocatalyst (Figure S6). The filtrates after the photocatalytic experiments were also subjected to ICP analysis and have shown that the concentration of Cd was around 18 mg/L, which represents dissolution of Cd. However, we are of the opinion that a significant contribution to this value is made by very fine CdS nanoparticles, which penetrated the filtration process and got into the filtrate. Thus, the amount of dissolved Cd ions is low.
I think there is an issue regarding the explanation of results. In the conclusion section, the authors stated that cCdS was more effective than aCdS because cCdS has a band gap higher than aCdS. However, this statement is not convincing. Usually, a lower band gap allows better light absorption and a higher yields of generated redox charges. I suggest to re-consider this statement.    We agree with this comment. This statement was modified. We have added some discussion how it is possible that material with higher bandgap shows better results. It therefore seems that not only bandgap value corresponds to the observed results, but there might be also other effect, like the presence of defects in mechanochemically treated solids, etc.

Reviewer 4 Report

In this manuscript, the CdS nanoparticles were prepared using two mechanochemical approaches. The synthesized “cCdS” and “nCdS” were accurately characterized by XRD, Raman, XPS, UV-vis, PL, TEM and DLS. However, the synthetic routes are not novel and the photocatalytic properties of cCdS are only slightly improved compared with nCdS. What’s more, the antibacterial performance of CdS nanoparticles is not very satisfactory. Therefore, it is not recommended this manuscript to be accepted for publication in its present form. Below are my general comments.

  1. Why does hydrogen production not increase linearly but decrease with time? (Figure 11) There may be some inaccuracies in the hydrogen generation experiments.
  2. The grammar and typos should be checked carefully. For example, it is "Na2S2O3•5H2O", not "Na2S2O3×5H2O".
  1. Which sample the “CdS” and “N-CdS” in Figure 12-14 and Figure S represent? The description should be expressed more clearly.
  2. Although the photocatalytic properties of cCdS are slightly enhanced compared with nCdS, the mechanism should be clarified.
  3. The SEM characterization of as-prepared samples should be provided.
  4. I think 93 references are too many for a research article.

Author Response

Reviewer’s comments Answers for comments
In this manuscript, the CdS nanoparticles were prepared using two mechanochemical approaches. The synthesized “cCdS” and “nCdS” were accurately characterized by XRD, Raman, XPS, UV-vis, PL, TEM and DLS. However, the synthetic routes are not novel and the photocatalytic properties of cCdS are only slightly improved compared with nCdS. What’s more, the antibacterial performance of CdS nanoparticles is not very satisfactory. Therefore, it is not recommended this manuscript to be accepted for publication in its present form. Below are my general comments.

We are surprised by the reviewer's opinion that the antimicrobial activity of tested materials was "not very satisfactory." Figure 14 clearly shows that in the first hours of culture, both CdS nanomaterials in the highest used concentration have considerable antimicrobial potential. Naturally, a bacterial population may regrow in the tested conditions. Nevertheless, we show that the antimicrobial effect is quite strong, and the effects of inhibiting the population growth can be seen even after 24 hours (Fig. 13). Nanomaterials act differently than antibiotics and are not directed into cells' physiology by specific interaction with critical biochemical pathways.

Furthermore, because of their size, the effects of nanomaterials are highly restricted by the availability of material. Once the material is covered with biomass, the rest of the population can go back to high growth. Even with antibiotics, the effectiveness is never 100%. Apart from availability, the occurrence of persister cells also may explain the population's survival. We have shown that the material can considerably inhibit their growth in conditions that are optimal for bacteria. In that view, the results seem satisfactory. 
Why does hydrogen production not increase linearly but decrease with time? (Figure 11) There may be some inaccuracies in the hydrogen generation experiments. In the text, the results and discussion of hydrogen generation was added an explanation about the difference in photocatalytic activity of cadmium sulfide nanoparticles obtained with different mechanochemical methods and reason for decrease of activity after 120 min. Decreasing of hydrogen generation can be caused by increasing of oxidation products of glycerol.
The grammar and typos should be checked carefully. For example, it is "Na2S2O3•5H2O", not "Na2S2O3×5H2O". We agree with the Reviewer. The grammar and chemical formula of crystalline hydrates in manuscript are corrected.
Which sample the “CdS” and “N-CdS” in Figure 12-14 and Figure S represent? The description should be expressed more clearly. We agree with the Reviewer. The description and name of the samples in the figures are corrected, we now use only aCdS and cCdS.
Although the photocatalytic properties of cCdS are slightly enhanced compared with nCdS, the mechanism should be clarified.

We agree that both types of nanoparticles have high photocatalytic activity, since the sizes of nanoparticles are very small, and the morphology is close. However, the difference in photocatalytic activity between cCdS and aCdS is noticeable in both hydrogen generation and Orange II decomposition. The photocatalytic hydrogen generation activity of cCdS is almost two times greater than aCdS. The reaction constant of the decomposition of Orange II on surface aCdS is k=0.012 min-1, and on surface cCdS is 1.5 times more and equal to k=0.018 min-1. In text was added explanation about potential mechanism. During the photocatalytic process, formed radicals, OH and O2-, decompose organic compound Orange II during photocatalytic process, by reacting to the aromatic ring of Orange II molecules and opening it at the azo bond and the hydroxylated ring, finally giving gaseous N2, H2O and CO2.

The SEM characterization of as-prepared samples should be provided. The detailed SEM analysis was performed, and the results are now included in the main body of the text (Fig. 7). The results of EDS analysis and elemental mapping are provided in the ESI (Fig. S4)

I think 93 references are too many for a research article.

We agree with the Reviewer comment. Some references were removed from the manuscript to decrease their amount.

Round 2

Reviewer 2 Report

I have no further comments, and the authors did not manage to response to most of the previous comments.

Author Response

Reviewer’s comments Answers for comments
I have no further comments, and the authors did not manage to response to most of the previous comments

The reviewers’ report submitted in round 1 contained 4 comments. We have to admit that our response to the first comment regarding the comparison with the performance of CdS nanoparticles prepared by other methods and/or commercial ones was insufficient. For this purpose, we have now included a table (Table 2) where we compared the photocatalytic activities of CdS nanoparticles prepared by our approach with that of CdS nanoparticles prepared by different types of synthesis methods. According to the collected data, it was found that our approach has a beneficial economic effect, since in 5 min we could obtain CdS nanoparticles with similar activity, thus saving time, environment and energy.

The second comment criticized the way how characterization results are presented, however, no details were provided. We have used the same way of results presentation that we have used in more papers before. According to us, the organization of the manuscript is fine, however, we are open to doing more changes if the reviewer will provide more specific comments.

In the 3rd comment, we really did not understand what the Reviewer had in my mind, how he wishes us to check the stability. We have asked him to be more specific in the second round, but he did not provide any details, and thus, it is impossible to answer this comment properly.

In the case of the last comment, we agreed with the Reviewer and deleted a couple of references (this was also the point of another reviewer).

Reviewer 4 Report

Acceptance.

Author Response

We thank the reviewer for the positive evaluation of our work!

Round 3

Reviewer 2 Report

I have no further comments.